# The Political Economy of Tobacco in Mozambique and Zimbabwe: A Triangulation Mixed Methods Protocol

**DOI:** 10.3390/ijerph17124262

**Published:** 2020-06-15

**Authors:** Raphael Lencucha, Jeffrey Drope, Ronald Labonte, Benedito Cunguara, Arne Ruckert, Zvikie Mlambo, Artwell Kadungure, Stella Bialous, Nhamo Nhamo

**Affiliations:** 1Faculty of Medicine, School of Physical and Occupational Therapy, McGill University, 3630 Promenade Sir William Osler, Montreal, QC H3G 1Y5, Canada; 2Economic and Health Policy Research, American Cancer Society, Atlanta, GA 30303, USA; jeffrey.drope@cancer.org; 3School of Epidemiology and Public Health, University of Ottawa, Ottawa, ON K1G 5Z3, Canada; rlabonte@uottawa.ca (R.L.); aruckert@uottawa.ca (A.R.); 4Independent Researcher, Av Vladmir Lenine #2081, Flat 1.4, Maputo P.O. Box 55, Mozambique; cunguara@gmail.com; 5Training and Research Support Center, Harare P.O. Box CY 2720, Zimbabwe; zvikie@tarsc.org (Z.M.); artwell@tarsc.org (A.K.); 6Social and Behavioral Sciences Department, School of Nursing, UCSF, San Franscisco, CA 94143, USA; Stella.Bialous@ucsf.edu; 7Institute of Research, Innovation and Technological Solutions, Zimbabwe Open University, Harare P.O. Box MP 1119, Zimbabwe; nnhamo@gmail.com

**Keywords:** tobacco control, agriculture, political economy, public policy

## Abstract

Changing global markets have generated a dramatic shift in tobacco consumption from high-income countries (HICs) to low- and middle-income countries (LMICs); by 2030, more than 80% of the disease burden from tobacco use will fall on LMICs. Propelling this shift, opponents of tobacco control have successfully asserted that tobacco is essential to the economic livelihoods of smallholder tobacco farmers and the economy of tobacco-growing countries. This nexus of economic, agricultural and public health policymaking is one of the greatest challenges facing tobacco control efforts, especially in LMICs. To date, there is a lack of comparative, individual level evidence about the actual livelihoods of tobacco-growing farmers and the political economic context driving tobacco production. This comparative evidence is critically important to identify similarities and differences across contexts and to provide local evidence to inform policies and institutional engagement. Our proposed four-year project will examine the economic situation of smallholder farmers in two major tobacco-growing LMICs—Mozambique and Zimbabwe—and the political economy shaping farmers’ livelihoods and tobacco control efforts. We will collect and analyze the existing data and policy literature on the political economy of tobacco in these two countries. We will also implement household-level economic surveys of nationally representative samples of farmers. The surveys will be complimented with focus group discussions with farmers across the major tobacco-growing regions. Finally, we will interview key informants in these countries in order to illuminate the policy context in which tobacco production is perpetuated. The team will develop country-level reports and policy briefs that will inform two sets of dissemination workshops in each country with relevant stakeholders. We will also conduct workshops to present our findings to the survey and focus group participants, and other members of these tobacco-growing communities, so they can directly benefit from the research to which they are contributing.

## 1. Introduction

Tobacco use remains the single most important cause of preventable morbidity and premature mortality worldwide. While 100 million people died from tobacco use in the 20th century, an estimated 1 billion people will die from tobacco use in the 21st century without effective policy interventions [1,2]. Importantly, 80% of these deaths will occur in low- and middle-income countries (LMICs) by 2030 [3]. Tobacco use, in particular, is rising rapidly in Sub-Saharan Africa (SSA) [2]. Tobacco control efforts worldwide are guided largely by the policy interventions elucidated in the Framework Convention on Tobacco Control (FCTC), an international treaty that currently has 181 parties (www.who.int/fctc/en/). The treaty identifies the importance of supply-side reduction (i.e., reducing tobacco production and availability), compelling FCTC parties to find sustainable alternative livelihoods for those working in the tobacco supply chain (Article 17). Moreover, the livelihood argument for tobacco farming (i.e., that it is essential for the economic development of tobacco-growing countries and the financial security of millions of smallholder tobacco farmers) is routinely invoked by the tobacco industry and used to gain access to, and influence, policymakers in key economic ministries in many LMICs [2,3,4]. There is little evidence about how and why farmers gain or lose from tobacco growing. However, the crude narrative of prosperity is proving successful in preventing the implementation of both demand- and supply-side tobacco control measures in most tobacco-producing countries [5,6,7,8]. We have found that the narrative that dominates in tobacco-growing countries is that tobacco production is an economic issue, and that it is primarily for export. However, the rising consumption rates in tobacco-growing countries adds weight to the need to explore how governments navigate the tension between economic and health policy.

This project will expand ongoing research that investigates how the political economy of tobacco supply affects tobacco control efforts in SSA (Kenya, Zambia, and Malawi) and Indonesia, and empirically examines the economic livelihoods of smallholder farmers. Importantly, we are also beginning to examine the mechanisms that perpetuate tobacco production and the policy and market levers that can create a shift towards alternatives. This study extends our research to two of the largest tobacco-producing SSA countries, Zimbabwe and Mozambique. Following the same two lines of inquiry as our ongoing research (described under Background and Rationale), this project will answer three overarching questions:1)What are the political and economic conditions that lead to policies in support of tobacco production and inhibit policies that support alternatives to tobacco growing (supply side)?2)How do these policy measures affect domestic tobacco control measures (demand side)?3)What are the actual economic livelihoods of smallholder tobacco farmers?

This proposal builds directly on two projects supported by the National Institutes for Health (NIH, USA)—Research and Capacity-Building at the Nexus of Tobacco Control and Economic Policymaking in Africa (2012–2017), and The Political Economy of Tobacco Farming in Low- and Middle-Income Countries (R01DA035158) (2017–2022)—as well as research funded by the Bloomberg Initiative to Reduce Tobacco Use conducted in the Philippines and Brazil from 2012–2015, and the World Bank conducted in Indonesia (2015–2017) [9]. These studies pursue two main lines of inquiry. The first line of inquiry explores the policy context with an emphasis on identifying the key conflicts that exist between health and economic sectors, and how to resolve these [10,11,12]. We examined opportunities within existing international economic structures to protect public health innovation [7,8,9]. We explored how tobacco-growing countries oppose tobacco control in key international economic fora, such as the World Trade Organization, using familiar tobacco industry arguments [5,6]. We also sought to explain how government institutions can shape this policy nexus [13], by incentivizing tobacco production [8], and how support for tobacco production has become institutionalized in tobacco-growing countries, such as the legislative protection of the tobacco industry in the Philippines [14] and the promotion of the industry in Zambia [8,15]. This line of inquiry has generated important findings on the factors that contribute to policy incoherence with respect to international health commitments (such as the FCTC) and trade commitments, as well as opportunities to enhance policy coherence in greater compliance with the FCTC [16]. Our studies in Kenya, for example, found that when the agriculture sector formed relationships with the health sector around tobacco control policies, they were more supportive of strategies to reduce tobacco production (e.g., [10,14]).

Our second line of inquiry involves an examination of the economic livelihoods of tobacco farmers. One of the key findings of this research is that, for governments in tobacco-growing countries, the core issue at the intersection of economic and tobacco control policies is the argument, actively promoted by the tobacco industry, that tobacco is a lucrative enterprise for smallholder farmers. Our inquiry included analyzing the livelihoods of tobacco farmers and the political economic context in which they produce tobacco for the market [17,18,19,20]. Until our detailed, statistically powered, and nationally representative farmer surveys, research on tobacco farmer livelihoods have most often been conducted with non-representative samples focusing on the broader challenges of tobacco farming [21], specific working conditions, including child labor [22,23,24] and mostly small, localized experiments with alternative crops [25,26]. Few studies had rigorously examined the economic status of a large, nationally representative sample of smallholder tobacco farmers [27,28], or systematically incorporated labor costs into the analysis [29]. The present studies also include variables that allow for an analysis of factors associated with tobacco growing, including proximity to markets, perceptions of viability and profitability, and access to credit and inputs (e.g., seed and fertilizer) [30].

## 2. Methods and Design

In a context of increasing regionalized trade treaties (and tobacco trade) in Africa, renewed emphasis on increasing economic development through agribusiness, and the intensity with which the tobacco industry is targeting Africa (for both tobacco production and consumption) [5,31,32,33,34,35], we will add Zimbabwe and Mozambique, the largest and fourth largest tobacco producing countries in Africa respectively, to our ongoing set of studies.

### 2.1. Ethics Approval and Consent to Participate

Institutional Review Board (IRB) approval was obtained from the Institutional Review Boards of the Faculty of Medicine at McGill University and the University of Ottawa. The team is in the process of applying for IRB approval in Zimbabwe and Mozambique. The application is being submitted to the Research Council of Zimbabwe (http://www.rcz.ac.zw/) and Zimbabwe Open University, as well as the Ministry of Health in Mozambique. Data collected from human subjects will require written consent.

### 2.2. Case Selection

Both Zimbabwe and Mozambique have seen a rise in tobacco consumption in the past four decades. Mozambique’s daily tobacco use rates are more than 30% for men and 9% for women [2,36]. Similarly, the daily smoking prevalence in Zimbabwe in 2015 was 31.2% for men and 2.1% for women, up from 22.9% for men and 0.5% for women in 2011 [37]. Particularly troubling is the higher rates of use among youth and those in the lowest economic quintile. These rates are alarming and reflect the general strategy of the tobacco industry to aggressively expand markets in LMICs. Zimbabwe and Mozambique were the latest (and close to the last remaining) two countries to sign the FCTC in 2014 and 2017, respectively. The early opposition to the FCTC from these two countries reflects the deep economic interests upholding tobacco growing within them. Both countries have implemented tobacco control measures in similar domains and to similar degrees, including taxation measures, smoke-free restrictions, and written (not graphic) health warnings, although both countries have yet to implement comprehensive tobacco control, as outlined in the provisions of the FCTC. Both countries also have major tobacco farming interests [38]. Zimbabwe is the largest producer of tobacco in Africa and is fifth in the world [39]; tobacco contributed about 11% to the country’s total gross domestic product (GDP) in 2014 [40] and over a quarter of the agricultural GDP [41]. There are roughly 170,000 registered tobacco farmers in Zimbabwe (2019 season), an increase of 46% from the previous season [42], about three quarters of which are smallholder farms [43]. Mozambique has also seen a huge rise in tobacco production [44] ranking fourth in Africa [39], with tobacco now one of Mozambique’s largest agricultural exports, accounting for roughly 34 percent of total agricultural exports and almost four percent of total exports of goods and services [45,46]. This rise is driven by policies implemented by both governments to facilitate the increase in tobacco production [41,47], even as the Zimbabwe agricultural investment plan notes that ‘the income from tobacco should be used to diversify to other products with comparative advantage’ [41], indicative of the common and persistent contradictions within government policy [8,14] and the importance of exploring these contradictions in depth in both of these countries.

### 2.3. Research Design

Our research utilizes a triangulation mixed method design [48]. This design enables our team to compare, contrast, and enrich findings using different methods. Thus, we can produce standalone findings using each method, and also cross-reference the findings to create a more nuanced analysis of the political economies that inhibit, or enable, effective tobacco control measures, including reductions in tobacco supply through alternative tobacco farmer livelihoods. With respect to this latter line of inquiry, our qualitative research with tobacco farmers can provide a more in-depth understanding of their livelihood concerns and constraints, contributing to the literature on the social context of tobacco growing, while also helping us to interpret generalized patterns and relationships identified through the survey research. By conducting a comprehensive policy review and key informant interviews at the policy level, we can link, for example, the variables associated with farmer decisions with policies that shape the landscape of the tobacco supply chain. This design is ideal to serve the two purposes of this research: (1) to inform tobacco control policy in the two studied countries, and (2) to produce academic and research-informed policy outputs that contribute to the wider discourse on controlling the tobacco supply. We have used this design in our past studies, with the ‘value added’ of the mixed methods being most apparent in the publicly available reports we have issued on each country’s findings [9,17,18,49].

### 2.4. Theoretical Framework

Our research uses a ‘3-i’ political economy theoretical framework, which focuses on the role of interests, institutions, and ideas in policymaking [50,51,52,53]. There are numerous interests affecting tobacco control and farming (e.g., industry, farmer organizations, and civil society) with legislative, regulatory and/or programmatic preferences. The tobacco industry seeks to ensure a steady supply of cheap tobacco leaf and that tobacco control efforts do not undermine their profit margins [54]. Tobacco farming organizations are meant to privilege the farmers’ needs, while research suggests that the tobacco industry often manipulates and/or funds such organizations to counter tobacco control efforts [55,56]. A small number of civil society organizations (CSOs) in both domestic and international arenas are actively promoting policies that would strengthen tobacco control and address tobacco farmers’ livelihoods, but not promote tobacco farming itself. These interests direct their policy preferences to formal governmental institutions (e.g., ministries of agriculture, trade and industry, finance, agrarian development, etc.) that not only mediate their diverse demands, but also develop independent (and often conflicting) policy preferences. Many institutional actors have longstanding close relationships with the tobacco industry and appear to privilege the industry’s preferences [8,15,57], which is further conditioned by ideas of economic development [16]. Ideas influence how different societal (institutional) actors define a problem, and how they perceive different policy options to be effective, feasible, and acceptable. As such, ideas influence agenda-setting, policy formulation, and implementation by determining which representations of the problem and policy solutions will be heard and understood by policymakers [58]. Over time, ideas become norms, and a collective understanding (idea) of what should be done (behavior) [59] develops, which, when widely accepted by actors, leads to mostly unquestioned assumptions that then dominate policy [60]. The 3-i political economy framework informs our policy document analyses and key informant interviews. It is also a useful heuristic for addressing research question 2 (i.e., How do these policy measures impact domestic tobacco control measures (demand side)?), and for a theoretical analysis of our research findings.

### 2.5. Data Collection

#### 2.5.1. Policy Document Analysis (Questions 1 and 2)

We will begin by collating and analyzing recent and current research, legislation, corresponding regulations, and reports pertaining to tobacco control and tobacco farming specific to Mozambique and Zimbabwe. Since the tobacco landscape is situated within national development policies, comparative historical analysis [61] will be used to track changes in policy over time (framing, rationale, and implementation) in the period leading up to the present study. Our in-country African partners will facilitate the collection of public records and, based on each country’s context, will advise on how far back a historical analysis should reach. The document collection and analysis will follow the protocol established by Arksey and O’Malley [62], which entails six stages, including identifying the research question (see above for our research questions), searching for relevant documents, selecting documents, charting data, collating, summarizing and reporting the results and conducting consultation exercises. Our search will be undertaken across sectors, including, but not limited to, agronomy research, taxation, extension services, and social support programs for farmers. We will also identify the types of relationships that exist between the health and economic sectors of government, and in what ways there is coherence and/or conflict among these policies. We will also track the discourses and ideas employed by key actors to foster particular policy choices (or to actively oppose others). Finally, we will examine the roles/actions/outputs of institutional actors, such as development agencies, multilateral institutions (e.g., World Bank, WHO, FCTC, etc.), the tobacco industry and others that may be influencing or have an interest in tobacco farming in these countries. This may involve tracing the influence of transnational and national tobacco companies over the years, a ‘periodization’ of tobacco industry development within the two countries. Recent research has illuminated the influence of China in shaping tobacco production in Zimbabwe [34,35]. The same can be explored in Mozambique, and by other tobacco companies in the two countries.

#### 2.5.2. Key Informant Interviews (KIIs) (Questions 1 and 2)

The research team will recruit key informants for in-depth interviews using purposive sampling [63]. The sample frame will replicate the one used in our three other SSA case study countries and will include individuals who meet one or more of the following criteria: representatives who have been actively involved in domestic tobacco policy including tobacco control and tobacco economics, agriculture and other production-oriented sectors, and major international economic and health policies (e.g., regional trade and investment agreements, the FCTC) or national development strategies. Individuals will initially be recruited from government (including health, trade, agriculture, finance and other relevant ministries), public health civil society organizations (CSOs), and the tobacco and allied industries. Because we are interested in the intersection of tobacco demand and supply policies, we will interview across sectors with one line of questions pertaining to coordination and cooperation across sectors. A snowball sampling technique will then be used to identify further participants who were not identified by our research team during the purposive recruitment period [64,65]. Our previous experience with Key Informant Interviews (KIIs) suggests that a sample target of 20 participants in each country will allow for a diverse and representative sample (i.e., different government sectors and agencies, civil society organizations and industry interests) and will be sufficient to gain a rich understanding of the policy context [66,67]. Although there is no accepted standard for determining sample size for qualitative KII studies, our minimum of *n* = 20 will ensure a rich and diverse data set based on the representativeness of organizations, agencies and sectors sampled, the anticipated length of the interviews, the specificity of the information sought and the triangulation methods used with the policy document analyses [68]. KIIs will be conducted in year 2, after completion of the policy document analysis, and repeated in year 4. We have found in our previous work that policy environments can change quickly, particularly given international pressures to reduce tobacco supply and parallel pressures to increase economic growth. Interviews will be semi-structured using a standard interview guide based on the one used in our NIH-funded projects. Interviews will be conducted face-to-face by at least two team members (one local, one international), at least one of whom will be a senior investigator, and will be audio-recorded and transcribed verbatim. Having two interviewers allows for consultation and deliberation before and after each interview pertaining to salient observations, guiding the analysis and facilitating a shared understanding of the research process over time.

#### 2.5.3. Tobacco Farmer Survey (Question 3)

We will undertake two waves of the panel survey in each country in years 1–2 and 4, creating a longitudinal design with replenishment in order to gain insights into any changes in tobacco production, including switching to other crops or other forms of employment and the factors informing such decisions. Wave 1 and 2 will take place in years 1–2 and year 4 (see Table 1). This time frame includes testing survey instruments on data collection devices and servers, hiring and training enumerators (*n* = ~8/country) and onsite data managers, travel to each tobacco-growing region, and the administration of the survey. The survey is administered in the form of a computer-assisted personal interview (CAPI) via tablet using a fillable form, which is automatically saved in a cloud storage system. The data quality is checked daily and revisits arranged if needed. The survey instrument was developed based on similar surveys in other countries and expanded by including items from the World Bank Living Standards Measurement Study (LSMS). The LSMS is a survey program that provides technical assistance to national statistical offices in designing and implementing multi-topic household surveys. The survey instrument is divided into 26 sections and includes the following topic headings: household characteristics; livelihood, income and assets; land ownership and crop production; tobacco production generally; tobacco production under contracts (where applicable); tobacco marketing; farmer debt or credit; household food security; the future of tobacco production; and health. Survey items were extensively field tested in the NIH-funded studies (three waves in four countries).

The survey will be loaded onto the tablets using CSPro (www.census.gov/data/software/cspro.html), and the data exported to Stata SE. CAPI helps to minimize human errors in data input, and we are able to record audio data, capture the GPS coordinates of the farms and photographs of contracts. We will select all of the major tobacco-growing regions in each country and conduct random sampling to construct a geographically representative sample of tobacco farmers. To decide on the minimum sample size of tobacco farmers, we adopt a conservative standard deviation of 0.5, a confidence level of 95% (Z = 1.96), Z being the desired confidence interval, and a margin of error of 5%. As the population sizes of farmers are large in the two countries, there is no further adjustment necessary to decide on the minimum sample size. We will aim for a sample size equal to or greater than 626. In the intended sampling process, we include (1) current and (2) former tobacco farmers, and (3) those living in the same geographical area who have never grown tobacco. The farmers who have never grown tobacco will provide important information about the economic viability, labor intensity and other factors associated with growing other crops in the same geographic and political economic context. This group provides an important point of comparison to understand differences and similarities in farmer household characteristics and decisions.

To maintain statistical power, a minimum distribution of 462 + 82 + 82 = 626 is required in Wave 1, assuming an attrition rate less than or equal to 20% based on similar surveys in neighboring countries. This sample provides the ability to achieve statistical analysis results to reflect the targeted study population while providing meaningful comparison groups for econometric models. Tobacco farmers are identified via farmer registries where available and by local leaders. Households are then randomly selected from a comprehensive list of all tobacco-growing households in a village. The survey administration begins in a focal village in each tobacco-growing region. We recruit farmers immediately following the end of the growing season to minimize recall bias. The enumerators typically spend 20 working days in the field administering the survey. Our country-level team members are experienced agricultural economists with extensive experience conducting large scale surveys with the World Bank, local governments and other institutions. Our multi-wave approach aims to ensure that we capture the complexities in changes in farm level practices and decisions. Our earlier research demonstrates that farmers tend not to move frequently, grow tobacco over many years, and be responsive to participation in surveys, so we anticipate low levels of attrition. However, attrition could potentially present a challenge to our study design. We will oversample by 20% in Wave 1 to account for possible attrition in Wave 2, based on actual attrition rates experienced between Waves 1 and 2 in our other SSA countries. We will monitor the characteristics of the dropouts and aim to replenish them with farmers similar in demographic characteristics. If attrition turns out to be a problem (after testing for attrition bias) even after oversampling farmers in Wave 1, we will generate inverse probability weights and apply those to the data, thus correcting for attrition bias.

#### 2.5.4. Tobacco Farmer Focus Group Discussions (FGDs) (Question 3)

As we have done previously, we will conduct three focus groups during each wave of the survey (six in total). Tobacco is mainly grown in eleven council districts in Zimbabwe and in three provinces in Mozambique [45,69]. Our focus groups will aim to capture diversity by being implemented in different tobacco-growing regions, emphasizing the three main tobacco-growing regions (i.e., those with highest number of farmers) in each country. For each focus group, we will seek a mix of gender, education level, and farming experience. Topics will include the history of farming in the area, seasonal and daily activities and schedules of household members, access to credit, debt, historical resource analysis, food security, gender issues, environmental factors, child labor and supply chain issues (e.g., transportation of leaves to market, satisfaction with tobacco price), and interaction with policy and policymakers (likely to be mainly local). Focus groups will be conducted by in-country team members with the goal of aligning the focus groups with the visit of the international researcher for the KIIs. The Focus Group Discussions (FGDs) will independently contribute to a deeper understanding of the issues facing tobacco farmers, while providing a qualitative basis to help interpret the survey findings.

#### 2.5.5. Participant and Public Involvement

The study design, as well as knowledge translation and dissemination activities, received input from stakeholders working in the health ministries of the two countries. Additionally, because this protocol draws from previous and ongoing research, the team was able to adapt aspects of the Knowledge Translation (KT) and dissemination activities based on previous experience with the reports, workshops and consultations within both the policy community and among farmers. Given that our research involves multiple waves of survey implementation, the participants have regular contact with the research team.

### 2.6. Data Analysis

Retrieved documents will be entered into Zotero reference management software. The included documents (e.g., academic articles, legislation and regulations, policy reports) will be imported into NVivo, a qualitative data analysis software, for coding. This software will help organize our analytical synthesis of extracted data. The first level of analysis will involve the establishment of key institutions and a timeline of key policies in each country. Secondly, we will conduct a constant comparative analysis to examine similarities and differences across the documents according to analytical categories, including policy solutions identified, governance mechanisms discussed, barriers and facilitators to addressing tobacco control and alternative tobacco farmer livelihoods, networks mentioned, and funding mechanisms. The analysis of policy documents will pay attention to the policy frames and evidence supporting the rationale for the policy measures. We will treat the review as an iterative process where we update the analysis as new documents are collected. This analysis will also allow us to construct a timeline of policies in relation to the broader political economy.

All KII and focus group transcripts will similarly be entered into NVivo and coded using a thematic analysis guided by our theoretical model. The 3-i framework will serve as the basic analytic framework guiding the data collection and analysis. For example, with ‘institutions’ in mind, we will code to develop an understanding of the types of rules, norms and strategies that shape the policy process. Following our practice in our other case study countries, a selection of three transcripts will initially be read by at least two team members, who will then discuss and reach agreement upon an initial coding tree for subsequent NVivo thematic analyses. Our qualitative analysis is informed by a constructivist grounded theory methodology, which aims to ‘construct’ an understanding of a particular process or phenomenon by engaging in an interpretative analysis of data [67]. The analysis involves coding the data to identify shared and differing perspectives on tobacco production and control and to construct a picture of the 3-i framework in each country. The coding process involves four steps. Open coding involves inductively naming segments of data. Focused coding involves synthesizing and grouping codes into larger categories or ‘meaning units’. Axial coding begins to tell a more complete story about the research question by making connections between categories, utilizing a coding paradigm involving conditions, context, action/interactional strategies and consequences [70]. Theoretical coding is the last step that involves developing a core category, systematically using sub-categories to conceptualize this core category, and filling in categories that need further refinement and development [67]. The research team will meet regularly to review emergent codes and return to original transcripts when disagreements over interpretations arise. Coding results produced by graduate students will also be discussed with the research team during regularly scheduled team meetings.

The survey data analysis plan will begin with descriptive statistics, using the most recent version of Stata SE (https://www.stata.com/products/which-stata-is-right-for-me/). Building on our previous and ongoing research, and using pertinent variables of all survey data observations, we will estimate two related but distinct types of profit. Firstly, for the farmers’ perceived profit, we will estimate average annual gross margins from tobacco-growing enterprises, which is the total revenue from tobacco sales minus all costs associated with growing tobacco, including all physical inputs (cost after deducting resell of equipment or recycled parts to other crops), fees, transportation, levies and hired labor. It does not include household labor. Secondly, we estimate a cost–profit calculation that incorporates the cost of a monetized value of household labor based on minimum wage measurements of agricultural day laborers in that specific region, which we argue is a conservative estimate of farmers’ wage value since they are typically more skilled than the workers that they hire. We define this enhanced measurement as the adjusted profits that the household earns from engaging in tobacco production, by accounting for the foregone labor earnings of household members in producing the tobacco crop. This is an accepted and increasingly used method in agricultural economics to estimate the opportunity costs of any agricultural activity [71,72,73].

We also focus on the dynamic of contracting. Smallholder tobacco farmers in Zimbabwe and Mozambique must choose whether to sign a contract to grow tobacco. This choice affects the level of interaction they have with tobacco firms, including access to inputs and markets, thereby possibly affecting profits [19,20]. Accordingly, we first compare perceived and adjusted average annual tobacco-specific profits between contract and independent farmers. We then further examine the social–economic factors associated with farming under a contract. Both complete case analysis and imputation for missing data using the hot deck nearest-neighbor method for all covariates will be adopted. The dependent variable, contract farming, is dichotomous. Using logistic regression, we will examine the association between contract farming and the social–economic characteristics of farmers. Additionally, random effect logistic regression models will control for possible regional differences. Covariates for analysis will draw from both previous agricultural research and our earlier research on tobacco farming, complemented by using a machine learning method, random forest [74,75,76], to ensure that we are not overlooking variables in our large dataset. Our survey also allows us to examine the factors that shape household decisions specific to female- versus male-headed households and, more robustly, the labor hours, roles, health indicators, education, etc. of females within male-headed households. This analysis contributes to our understanding of how gender dynamics shape household farming practices and decisions [77]. Finally, we will analyze the survey data to determine the reasons given for growing tobacco and the socio-economic conditions associated with the decision to continue tobacco production, and for farmers in Wave 2 who decided to stop, or to continue, growing tobacco.

#### Knowledge Translation (KT) and Dissemination

Our approach to KT will combine elements from end of grant KT and integrated KT. Our outputs from the grant will include traditional academic dissemination activities, including research articles and conference presentations, and a media engagement strategy to publicize our findings (including a series of webinars). We will host two types of dissemination workshops: one with policy stakeholders and the other with tobacco farmers. We will develop country reports (~30 pages × 2) and a two- to four-page companion policy brief in year 2 using the analysis of our initial findings from Wave 1 of the survey, KIIs and focus groups. This brief will be compared/contrasted with survey results in earlier projects and shared through workshops in each country. As we have done in the earlier project, we will disseminate the results to the tobacco farmers. Both sets of workshops involve an oral presentation of the findings with visual aids followed by a discussion. We have typically worked with farmer organizations to present our results to the survey participants and their peers and to generate more discussion about livelihoods. The second policy reports and companion policy briefs will incorporate any updates/changes based on Wave 2 survey results, focus groups and KIIs. The final reports will be published on the American Cancer Society website, the Tobacco Atlas website, and the websites of all affiliated universities and supportive research organizations. The second set of dissemination workshops will be hosted at the end of year 4. Our policy-relevant messages will be simple, action-oriented, and tailored for each audience to ensure that the information is user friendly and actionable. We will rely upon our two key knowledge users, each of whom serves as the focal point for tobacco control within the Ministry of Health, to serve as intermediaries who can enhance collaborations between researchers and knowledge users, effectively disseminate evidence to shape decisions, and assess, interpret, and adapt evidence to local context. The dissemination workshops will be hosted in collaboration with our two knowledge users, local research team members and one international team. The webinars will be part of a series of webinars offered through the Tobacco Atlas website (https://tobaccoatlas.org/), co-hosted by the American Cancer Society and Vital Strategies.

## 3. Conclusions

Strengths and Limitations of this Study

(a)The survey-based evaluation of livelihoods of tobacco farmers in comparison to former and never tobacco farmers is novel in the field of tobacco control;(b)The mixed methods design provides an opportunity to situate economic and other aspects of tobacco farming in the context of the broader political economies of Mozambique and Zimbabwe;(c)The inclusion of household labour, gender and other variables provides a unique lens to examine the profitability of tobacco farming and other social and economic impacts on households;(d)The recruitment of key informants to understand the policy context, institutional environment and policy processes may be constrained by the various interests involved in supporting tobacco growing as an economic development strategy;(e)The dissemination and knowledge translation approach may help inform both farmer decision making and public policy in a field that often lacks locally generated information.

## Figures and Tables

**Table 1 ijerph-17-04262-t001:** Timeline of research activities.

Research Activities	Year 1	Year 2	Year 3	Year 4
IRB Application																
Policy Document Analysis																
Farmer Survey																
Focus Groups																
Ethnographic study																
KIIs																
Policy workshop																
Policy report																
Farmer workshop																
Team meeting

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
