# Peer review of "The Political Economy of Tobacco in Mozambique and Zimbabwe: A Triangulation Mixed Methods Protocol"

_ijerph, 2020, doi:10.3390/ijerph17124262_

Round 1

Reviewer 1 Report

This is a rather well designed mixed-method study protocol aiming to explore relevant TC research gaps in low-income SSA countries.

The detail of the study is exhaustive and the manuscript reads very well.

My only suggestion is to adapt the abstract to better highlight the innovation of the study; and explain that context matters in policy and economy, pointing to the need to conduct this study in other African countries.

Author Response

Thank you for this positive feedback and this suggestion. We have added a sentence to the abstract to better illustrate why this research is important: “This comparative evidence is critically important to identify similarities and differences across contexts and to provide local evidence to inform policy and institutional engagement.”

Reviewer 2 Report

Overall the protocol seems important and scientifically strong; however, ethical concerns were raised because there appeared to be a lack of collaboration with institutional review boards or equivalents in each of the target countries. Existing plans should be clearly articulated for the reader. Otherwise plans should be developed and included in the protocol.

Below are sample resources for consideration.

Sample Resources:

https://www.hsph.harvard.edu/region-map/research_project/zimbabwe/

https://www.hsph.harvard.edu/region-map/research_project/mozambique/

Author Response

Thank you for this feedback and for these two resources. We hadn’t realized that we did not describe the process we are going through at the moment to obtain ethics approval in the host countries. We have added this description on page 3 in the section on Ethics Approval and Consent to Participate. Thank you for flagging this oversight.

Reviewer 3 Report

The paper consists in research protocol of the study with funding from the Canadian Institute of Health Research  which was obtained through a competitive peer review process.  Therefore, conflict of interest cannot be suspected. On the other hand, the protocol -  as I read it -  does not explore any questions specific to involvement of foreign tobacco companies, including Canadian and the US-based companies in building and sustaining tobacco industries in Mozambique and Zimbabwe. Absence of this question in the protocol constitutes substantial deficiency and needs to be ameliorated by presenting international context of the tobacco policies, including such questions as  profits of international companies from both countries and perception of external pressures by national and local stakeholders.

Despite this fundamental problem of the paper I have a number of more detailed questions.

In the beginning of the introduction in is predicted that in the 21st century one billion people die from tobacco use of which 80% in LMICs by 2030 (lines 50-52). This would imply that 800 million of this people die in LMICs until 2030 while remaining 200 million in next 70 years globally which is very improbable prediction, no matter of its source.

I presume that NIH (line 77) refer to the US institutes but it should be clearly stated as NIH exist in a number of countries.

Lines 122-124 claim that Zimbabwe and Mozambique have seen a massive rise in tobacco consumption in the past four decades but this is supported in the subsequent lines by the data covering four years 2011-2015.

In theoretical framework, a role of foreign tobacco companies is missing  while they have their vested interests, institutions and ideas.

In the section on tobacco farmer survey provision of the data how big is their population and what is their proportion among all farmers would be appropriate to justify calculation of the sample size. Moreover, data of a proportion of female farmers would be important not only to calculate their share in the sample of farmers but also to understand better cultures of tobacco farming and related economies in both countries.

Author Response

Comments and Suggestions for Authors

The paper consists in research protocol of the study with funding from the Canadian Institute of Health Research  which was obtained through a competitive peer review process.  Therefore, conflict of interest cannot be suspected. On the other hand, the protocol -  as I read it -  does not explore any questions specific to involvement of foreign tobacco companies, including Canadian and the US-based companies in building and sustaining tobacco industries in Mozambique and Zimbabwe. Absence of this question in the protocol constitutes substantial deficiency and needs to be ameliorated by presenting international context of the tobacco policies, including such questions as  profits of international companies from both countries and perception of external pressures by national and local stakeholders.

This is a fair point. We had alluded to the exploration of the impact and dynamics of tobacco companies in the two countries, but we have made this more explicit in the following statement added on line 218: “This may involve tracing the influence of transnational and national tobacco companies over the years, a ‘periodization’ of tobacco industry development within the two countries. Recent research has illuminated the influence of China in shaping tobacco production in Zimbabwe [34-35]. The same can be explored in Mozambique, and by other tobacco companies in the two countries.”

Despite this fundamental problem of the paper I have a number of more detailed questions.

In the beginning of the introduction in is predicted that in the 21st century one billion people die from tobacco use of which 80% in LMICs by 2030 (lines 50-52). This would imply that 800 million of this people die in LMICs until 2030 while remaining 200 million in next 70 years globally which is very improbable prediction, no matter of its source.

Thanks for pointing this out. Yes, this would be improbable. However, the statement doesn’t differentiate between the decades within the century, but rather estimates that if tobacco control measures are not implemented then 800 million will die from tobacco-related deaths in LMICs between 2000-2100.

I presume that NIH (line 77) refer to the US institutes but it should be clearly stated as NIH exist in a number of countries.

Thanks for this point. We have specified that these projects were funded through NIH, the US agency.

Lines 122-124 claim that Zimbabwe and Mozambique have seen a massive rise in tobacco consumption in the past four decades but this is supported in the subsequent lines by the data covering four years 2011-2015.

We have taken out ‘massive’ which is subjective. The following figures provide specific numbers for this rise in consumption rates.

In theoretical framework, a role of foreign tobacco companies is missing  while they have their vested interests, institutions and ideas.

See above

In the section on tobacco farmer survey provision of the data how big is their population and what is their proportion among all farmers would be appropriate to justify calculation of the sample size. Moreover, data of a proportion of female farmers would be important not only to calculate their share in the sample of farmers but also to understand better cultures of tobacco farming and related economies in both countries.

This is a really good point. We have decided to pursue the gendered dynamics primarily through focus groups, but we will use survey data from female headed households to complement the focus group data. Given that our main emphasis is on contract versus independent, tobacco versus non- or former-tobacco farmer, we have decided to keep to the sampling strategy articulated in this proposal.

Reviewer 4 Report

The paper proposes a new protocol (Triangulation Mixed Methods) to evaluate the political Economy of Tobacco in two major tobacco-growing LMICs (Mozambique e Zimbabwe).
The scientific sound is good.
I suggest to the Authors to explain better the overarching questions a (supply-side) and b (demand-side). Moreover the paragraph 2 have to be reorganized to improve readability. For example the sequence of paragraphs could be: 2.1; 2.3; 2.4; 2.2; 2.5 and 2.6.
At row 387 the number of the paragraph is 2.6.1.
The conclusions have to be improved.

Author Response

The paper proposes a new protocol (Triangulation Mixed Methods) to evaluate the political Economy of Tobacco in two major tobacco-growing LMICs (Mozambique e Zimbabwe).

The scientific sound is good.

Many thanks.

I suggest to the Authors to explain better the overarching questions a (supply-side) and b (demand-side). Moreover the paragraph 2 have to be reorganized to improve readability. For example the sequence of paragraphs could be: 2.1; 2.3; 2.4; 2.2; 2.5 and 2.6.
At row 387 the number of the paragraph is 2.6.1.

Thank you for these suggestions. We have added more on the supply-demand dynamic in the introduction (lines 69-72). We have also edited the mistake on line 403 (formerly 387). We have chosen to keep the order of sub-sections in section 2. We hope that is okay.

The conclusions have to be improved.

We have tried to make pointed conclusions. We hope that it is okay to maintain the conclusion section as is, unless there is specific suggestions to improve this final section.